# MERTK^+/hi^ M2c Macrophages Induced by Baicalin Alleviate Non-Alcoholic Fatty Liver Disease

**DOI:** 10.3390/ijms221910604

**Published:** 2021-09-30

**Authors:** Yin-Siew Lai, Huyen Thi Nguyen, Farrah P. Salmanida, Ko-Tung Chang

**Affiliations:** 1Department of Biological Science and Technology, National Pingtung University of Science and Technology, Pingtung 91201, Taiwan; juniorsu98@gmail.com (J.); Huyennguyendhv94@gmail.com (H.T.N.); fapsa98@gmail.com (F.P.S.); 2Research Center for Animal Biologics, National Pingtung University of Science and Technology, Pingtung 91201, Taiwan; rindy.0802@gmail.com; 3Flow Cytometry Center, Precision Instruments Center, National Pingtung University of Science and Technology, Pingtung 91201, Taiwan

**Keywords:** NAFLD, baicalin, macrophage, M2c, MERTK

## Abstract

Nonalcoholic fatty liver disease (NAFLD) is one of the most common liver diseases worldwide. An accumulation of fat, followed by inflammation, is the major cause of NAFLD progression. During inflammation, macrophages are the most abundant immune cells recruited to the site of injury. Macrophages are classified into “proinflammatory” M1 macrophages, and “anti-inflammatory” M2 macrophages. In NAFLD, M1 macrophages are the most prominent macrophages that lead to an excessive inflammatory response. Previously, we found that baicalin could polarize macrophages into anti-inflammatory M2c subtype macrophages with an increased level of MERTK expression. Several studies have also shown a strong correlation between MERTK expression and cholesterol efflux, efferocytosis, as well as phagocytosis capability. Therefore, in this study, we aim to elucidate the potential and efficacy of mononuclear-cell (MNC)-derived MERTK^+/hi^ M2c macrophages induced by baicalin as a cell-based therapy for NAFLD treatment. In our results, we have demonstrated that a MERTK^+/hi^ M2c macrophage injection to NAFLD mice contributes to an increased level of serum HDL secretion in the liver, a decline in the circulating CD4^+^CD25^−^ and CD8^+^CD25^−^ T cells and lowers the total NAFLD pathological score by lessening the inflammation, necrosis, and fibrosis. In the liver, profibrotic *COL1A1* and *FN**,* proinflammation *TNFα**,* as well as the regulator of lipid metabolism *PPARɣ* expression, were also downregulated after injection. In parallel, the transcriptomic profiles of the injected MERTK^+/hi^ M2c macrophages showed that the various genes directly or indirectly involved in NAFLD progression (e.g., *SERPINE1*, *FADS2*) were also suppressed. Downregulation of cytokines and inflammation-associated genes, such as *CCR5*, may promote a pro-resolving milieu in the NAFLD liver. Altogether, cell-based therapy using MERTK^+/hi^ M2c macrophages is promising, as it ameliorates NAFLD in mice.

## 1. Introduction

Nonalcoholic fatty liver disease (NAFLD) encompasses a broad spectrum of liver conditions varying in the severity of injury resulting from nonalcoholic consumption. Further classification has divided NAFLD into two categories, nonalcoholic fatty liver (NAFL) and nonalcoholic steatohepatitis (NASH). The NAFL is defined as simple hepatic steatosis, while NASH is characterized by a more serious hepatic steatosis with the addition of inflammation and fibrosis, which may lead to a more severe condition called cirrhosis [1,2,3]. In the last few years, the clinical and economic burden of NAFLD has become obvious and is expected to climb steeply in the coming decades because of the increased prevalence of type 2 diabetes and obesity [4]. Recently, nonalcoholic steatohepatitis (NASH) was found to be correlated with cardiovascular disease (CVD) events independently [5]. To date, the worldwide prevalence rate of NAFLD in the general population is estimated to be at 25%, and its annual toll is only expected to increase every year [6].

The hepatocellular accumulation of lipids, followed by inflammatory cell infiltration, are the hallmarks of steatohepatitis in NAFLD. Among inflammatory cells, macrophages serve as the first line of defense against infections and tissue injury [7]. Macrophages are highly plastic cells [8], of which classical M1 macrophages exert proinflammatory activity by releasing proinflammatory cytokines, such as IL-1β, IL-6, IL-8, IL-12, and TNF*α,* which could lead to tissue injury. On the contrary, their counterpart, M2 macrophages, promote anti-inflammatory features and are mainly involved in wound healing, tissue repair, and phagocytic clearance of cellular debris by releasing Arginase-1, IL-10, TGF-β, and matrix metalloproteases [9,10]. Furthermore, M2 macrophages can be subdivided into several types based on their stimuli: M2a (stimulated by IL-4 and IL-13); M2b (induced by combined immune complexes and toll-like receptor agonists); M2c (activated by IL-10, TGF-β, and glucocorticoids); and M2d (elicited by IL-6, TLR ligands, and adenosine) [11,12,13]. M2a macrophages are wound-healing macrophages that express high levels of mannose receptor (MR, also called CD206), secrete profibrotic factors, such as TGF-β, insulin-like growth factor (IGF), and fibronectin, and contribute to tissue repair. M2b macrophages possess both protective and pathogenic roles which secrete both pro- and anti-inflammatory cytokines. The M2c phenotype displays a regulatory phenotype and can repress inflammation and fibrosis and promote tissue repair. In addition, M2c macrophages are involved in the phagocytosis and efferocytosis of apoptotic cells via the upregulation of MERTK receptor [14]. M2d macrophages resemble tumor-associated macrophages (TAM), and mainly contributing to angiogenesis and metastasis and support tumor growth [12]. In NAFLD progression, the hepatic macrophage pool orchestrates a lot of crosstalk among resident or infiltrated cells, thus driving inflammatory processes. These aforementioned interactions trigger M1 macrophage activation, leading to tissue injury [15]. Therefore, M2 macrophage activation, especially the M2c subtype, is preferable due to its crucial role in creating an anti-inflammatory milieu and provoking tissue repair.

Baicalin (7-glucuronic acid, 5, 6-dihydroxyflavone), a major bioactive flavonoid compound found in the dried root of *Scutellaria baicalensis* Georgi (SBG), has been used and studied intensively in traditional Chinese medicine for its therapeutic potentials and broad pharmacological properties, including anti-inflammation, antioxidant, antiviral, and anticancer properties [16,17,18,19]. A recent study has shown that baicalin is capable of repolarizing LPS-induced M1 macrophages into the M2 phenotype characterized by the downregulation of *IRF5*, *TNFα*, and *IL-23* in vitro [20]. Furthermore, our recent study demonstrated that baicalin-induced M2c macrophage polarization is proven by the high expression of MERTK, interferon regulatory factor 4 (*IRF4*), interleukin-10 (*IL-10*), and *PTX3*. Interestingly, the increased expression of MERTK in macrophages has a strong correlation with both phagocytosis and efferocytosis activity [14].

Mer tyrosine kinase (MERTK), a member of the family of Tyro3/Axl/Mer (TAM) receptor kinases, is particularly involved in the mediation of early apoptotic cell clearance, immune regulation, platelet aggregation, cell proliferation, and proinflammatory cytokines suppression [21,22]. In addition, MERTK signaling promotes the expression and secretion of immune suppressive cytokines (IL-10, IL-13, IL-4, and TGFβ1) and represses proinflammatory cytokines (IL-12, IFN). Hence, MERTK^+^ cells promote immune tolerance and tissue repair. MERTK expression is mainly found on the surface of macrophages and dendritic cells, while the latter preferentially rely on Axl and Tyro3 [23,24]. The physiological roles of MERTK are acquired when the “bridging ligands” of the MERTK receptor (Gas6 and ProS) interacts with the “eat me” signals of phosphatidylserine (PtdSer) that present on apoptotic cell surfaces [25,26]. Once MERTK is activated, it triggers the synthesis of inflammation resolution mediators that counterbalance the proinflammatory mediators during acute inflammation through the suppression of calcium/calmodulin-dependent protein kinase II (CaMKII) activity [27]. In liver inflammation, macrophages accumulating around the necrosis area have orchestrated both tissue-destructive and resolution/repair responses [28].

Studies in humans and mice have shown the tendency of macrophages expressing MERTK to execute enhanced reparative responses [22,25,26]. By polarizing macrophages into the M2c subtype with a high expression of MERTK receptor, not only the high efficiency of efferocytosis and phagocytosis could be attained, but also could enhance the anti-inflammatory properties and cholesterol efflux [24,29]. Based on the fact that NAFLD has a strong correlation with excess inflammation response and impaired cholesterol efflux capacity (CEC), we further elucidate the potential and efficacy of baicalin-induced mononuclear-cell (MNC)-derived MERTK^+/hi^ M2c macrophages to be used as a cell-based therapy to treat NAFLD. We designed diseased mouse models of atherogenic-diet-induced NAFLD for cellular immunotherapy, and we provide a transcriptomic analysis profile to further elucidate the pathways underlying this phenomenon.

## 2. Results

### 2.1. M2 Macrophages Express High MERTK in the Presence of Baicalin

In order to generate MERTK-high-expressed M2c macrophages, we first collected mononuclear cells from the bone marrow of the mice and differentiated them into M2 bone-marrow-derived macrophages (BMDMs) by the induction of M-CSF for seven days followed by the addition of baicalin for 24 h to induce M2c subtype macrophages (hereafter named M2 and M2c macrophages, respectively). In parallel to our preliminary results [14], the polarization of M2c macrophages with baicalin show a specific round shape compared to that with M-CSF-treatment alone (Figure 1A). Next, to confirm whether BMDMs were polarized into M2c macrophages, we characterized them by using M2 macrophage protein surface marker CD11b. Flow cytometry data demonstrated that 84.3% of BMDMs were CD11b^+^/MERTK^+^ cells, while treatment with baicalin could further elevate the population of the cells that expressed both these markers up to 90% (Figure 1B,C). Moreover, MERTK surface proteins were ubiquitously expressed on BMDMs as proven by the mean fluorescence intensity (MFI). Polarization with baicalin further increases the MFI of MERTK compared to the treatment with M-CSF alone (Figure 1D). Thus, based on the data provided, MERTK^+/hi^ M2c macrophages as definition were successfully generated.

### 2.2. Injection of MERTK^+/hi^ M2c Macrophages Increases Serum High-Density Lipoprotein and T Cell Immunomodulation in the Peripheral Blood

In this study, an NAFLD mouse model was generated by feeding an atherogenic diet for seven weeks (Figure 2A). An atherogenic diet has been proven to induce fatty liver because of its high cholesterol content [30,31]. Next, the mice were divided into three groups: the normal chow diet group, the atherogenic diet group, and the atherogenic diet with a 100 μL of 10^6^ MERTK^+/hi^ M2c macrophage injection. Intravenous injection was subjected via retro-orbitally from week four to week seven (Figure 2A). Intravenous injection was chosen to effectively deliver the macrophages to the target site [32,33]. The body weight of the mice was weighed and compared to the control group every week. As expected, the weight of the mice fed with an atherogenic diet significantly increased as early as the second week of feeding compared to the normal chow group (Figure 2B). While in the treatment group with the injection of MERTK^+/hi^ M2c macrophages, the body weight was reduced significantly compared with the mice that received an atherogenic diet alone in the fifth week and seventh week. The mice that received an atherogenic diet showed an overall increase in body weight compared to the treatment group. To validate the efficacy of MERTK^+/hi^ M2c macrophage treatment, the mice were sacrificed after seven weeks of an atherogenic diet for further analysis. We found that the livers (Figure 2C) from normal chow diet mice were reddish with smooth and shiny phenotypes. On the contrary, the livers from the NAFLD group that received the continuous administration of an atherogenic diet for seven weeks were extremely pale, enlarged, and extensively infiltrated with white spots, which resemble hepatic fat accumulation (steatosis), as well as reddish spots and streaks that resemble liver hemorrhage (Figure 2C, indicated with the white arrow). Interestingly, the treatment group, compared to the NAFLD group, also showed pale livers in the presence of fat accumulation, but little to no hemorrhage was observed. Next, we tried to weigh the livers in order to observe whether the atherogenic diet could successfully increase liver weight by the accumulation of hepatic fat. Compared to the normal liver, both the NAFLD group and the treatment group displayed a significant increase in liver weight (Figure 2C). However, the treatment group showed a slight decrease in overall liver weight compared to the atherogenic diet group, albeit not significant.

In NAFLD patients, the liver overproduces several atherogenic factors, such as cytokines and “bad” lipoproteins. In this manner, fatty liver is associated with decreased serum high-density lipoproteins (HDL), combined with increased low-density lipoproteins (LDL), very-low-density lipoproteins (vLDL), and triglycerides that represent a threat for chronic inflammation-related diseases, such as NAFLD and CVD development [34]. On the basis of the fact that lipoprotein has a strong correlation with NAFLD progression and, counterintuitively that the treatment of MERTK^+/hi^ M2c macrophages could reduce the level of “bad” lipoprotein, we measured the levels of the plasma lipids of each group. Next, we collected the blood from the venous sinus (retro-orbital plexus), and the blood plasma was subjected to plasma lipid measurements (total cholesterol, vLDL/ LDL, and HDL). Indeed, the mice that received an atherogenic diet exhibited significant increases in plasma levels of total cholesterol and vLDL/LDL levels, but not HDL (Figure 2D). While in the group treated with with MERTK^+/hi^ M2c macrophages, the total cholesterol, and the levels of vLDL/LDL increased significantly compared to the control group. Surprisingly, the mice that received MERTK^+/hi^ M2c macrophages showed significant elevation in high-density lipoprotein (HDL) cholesterol, a negative predictor of NAFLD, although in wild mouse strains, such as C57BL6, the majority of cholesterol is carried in HDL [35]. From the results obtained, we concluded that the treatment of NAFLD with MERTK^+/hi^ M2c macrophages can modulate lipid homeostasis by increasing the level of HDL secretion in the liver, which could counter the bad influence of vLDL/LDL.

Pathophysiologically, the immune system itself is one of the major drivers of NAFLD progression and other obesity-associated comorbidities, and both the adaptive and innate immune systems are involved [36,37]. As one of the major antigen-presenting cells (APCs), and as part of the innate immune system, macrophages, especially the M1 phenotypes, are capable of secreting various proinflammatory cytokines and controlling the activation of the adaptive immune system, such as T cells and their effector phenotypes. The activation of T cells usually occurs in the spleen and is presented in the peripheral blood as mature circulating lymphocytes [38]. In NAFLD, it has been shown that Treg cells appear to have an overall tempering effect, while CD4^+^ and CD8^+^ T cells were more likely to induce liver damage and fibrosis progression, aggravating the burden of NAFLD and inflammation [36]. Therefore, we tried to analyze the immune cell population in the peripheral blood in all groups by using the surface protein marker CD4^+^CD25^−^ (helper T cells), CD8^+^CD25^−^ (cytotoxic T cells), and CD4^+^CD25^+^ (regulatory T Cells). As a result, the flow cytometry analysis strongly demonstrated that the group that received an atherogenic diet had an increased population of helper CD4^+^CD25^−^ T cells, as well as cytotoxic CD8^+^CD25^−^ T cells, in the peripheral blood compared to the control group (Figure 2E), which exacerbate liver injury. As expected, a group of CD4^+^CD25^+^ regulatory T cells was also increased as a response of the body to suppress excess T cell infiltration. Meanwhile, the treatment group showed a decreased population of overall circulating T Cells in the peripheral blood, suggesting lower T cell infiltration to the liver and, hence, improving liver condition. From these data, we conclude that treatment with MERTK^+/hi^ M2c macrophages stimulates the immunomodulation of T cells.

### 2.3. Injection of MERTK^+/hi^ M2c Macrophages Alleviates Atherogenic-Diet-Induced NAFLD In Vivo

In addition, to get a better comprehension of how MERTK^+/hi^ M2c macrophages affect the atherogenic-diet-induced NAFLD liver, a histology examination was performed by fixing the tissue into paraffin and then sectioning it into approximately 5–10 µm thicknesses. All the tissue sections were subjected to H&E as well as modified Masson’s trichrome staining. NAFLD has been associated with the abnormal accumulation of fat within hepatocytes (steatosis), swelling, and the rounding up of hepatocytes (hepatocyte ballooning), as well as inflammation. Moreover, the progression of NAFLD to the NASH stage is usually indicated by the increased percentage of overall steatosis and fibrosis [39,40]. A histopathological examination of the liver demonstrated that, compared to the control group, which has a normal appearance, the atherogenic diet group showed steatosis with the irregular shape of ballooned hepatocytes. Multiple focal randomly-distributed aggregates of a heterogenous population of inflammatory cells (foci), which resemble lipogranuloma/microgranuloma, were also observed around the portal (periportal inflammation) and the hepatic lobule (lobular inflammation) (Figure 3). As expected, NAFLD mice treated with MERTK^+/hi^ M2c macrophages significantly alleviated liver burden indexed by the reduction of inflammation and hepatic steatosis, even though hepatocytes ballooning was still present. Moreover, modified Masson’s trichome staining was used to determine the fibrosis area in the liver because of its nature to stain connective tissue. It has been frequently used in the clinical settings. Masson’s trichrome staining revealed fibrosis events across the liver section, especially around the portal vein (PV) and the sinusoidal area (Figure 4). Compared to the atherogenic diet group, treatment with MERTK^+/hi^ M2c macrophages reduced the prevalence of fibrosis in the portal/periportal area, and moderately restricted pericellular fibrosis around the liver bile canaliculi (Table 1). Overall, the presented data suggest that MERTK^+/hi^ M2c macrophages may play a beneficial role in NAFLD.

### 2.4. MERTK^+/hi^ M2c Macrophages Suppress the Proinflammatory TNFα, Profibrotic COL1A1, FN, and Lipid Regulator PPARɣ Expression in the Liver of NAFLD

Progression of NAFLD has been strongly associated with fibrosis and metabolic dysfunction [41]. To further validate our result, hepatic mRNA expression related to fibrosis, inflammation, and lipid metabolic was analyzed. To elucidate whether the MERTK^+/hi^ M2c macrophages reached the liver, we first analyzed the relative gene expression of MERTK in the liver. As expected, the treatment group displayed higher MERTK relative gene expression compared to both groups, suggesting the presence of MERTK^+/hi^ M2c macrophages in the liver. Furthermore, the hepatic mRNA level of collagen type 1 alpha (*COL1A1*), fibronectin (*FN*), peroxisome proliferator-activated receptor-gamma (*PPARɣ*), and tumor necrosis factor-alpha (*TNFα*) were measured using qPCR to assess the progression of liver damage and function. As shown in Figure 5, treatment with MERTK^+/hi^ M2c macrophages downregulates the relative expression of *COL1A1* and *FN* compared with that in the Ath diet group (*p* < 0.05), suggesting a decrease in overall fibrogenesis events. Furthermore, the regulator of lipid metabolism, *PPARɣ*, was also significantly downregulated in the treatment group compared with the Ath diet group, suggesting that fat accumulation and inflammation were reduced significantly. In addition, a significant decrease in the expression of inflammation-associated cytokine *TNFα* was also observed in the treatment group compared with that in the atherogenic diet group, suggesting lower hepatic inflammation. Herein, we again demonstrated that MERTK^+/hi^ M2c macrophages may contribute to the alleviation of Ath-diet-induced NAFLD by decreasing the levels of *COL1Al**, FN,* and *PPARɣ*.

### 2.5. Transcriptomic Analysis of MERTK^+/hi^ M2c Macrophages in the Alleviation of NAFLD

The introduction of the MERTK^+/hi^ M2c macrophages to the atherogenic-diet-induced NAFLD mice model have been shown to alleviate the fatty liver burden. This phenomenon leads us to a question: how and why does the injection of MERTK^+/hi^ M2c macrophages alleviate fatty liver burden? To answer these questions, we tried to gain insight into the transcriptomic regulation by comparing MNC-derived BMDMs, with and without baicalin induction, using next-generation sequencing (NGS). A differential expression analysis (DEA) revealed 221 differentially expressed genes (DEGs) when the transcriptomes of the M2 and MERTK^+/hi^ M2c macrophages were compared (41 upregulated and 180 downregulated; false discovery rate [FDR] ≤ 0.05, Figure 6A). Among these differentially expressed genes (DEGs), we focused on the top 40 upregulated and downregulated genes with at least a two-fold increase or a two-fold decrease (log2 fold change > 1 or log2 fold change < −1) (Figure 6B,C). In particular, MERTK^+/hi^ M2c macrophages exhibited a lower expression of genes that encode for proteins associated with cell cycles (e.g., *BARD1**, CENPF*, *BIRC5**, MYBL2, FOXM1*, *MELK**,* and *AURKB*). Additionally, genes involved directly or indirectly in NAFLD progression (e.g., *SERPINE1*, *FADS2*) were also downregulated. On the other hand, baicalin-induced MERTK^+/hi^ M2c macrophages also demonstrated a higher expression of genes associated with several functions in the cellular response to vascular endothelial growth factor (*NR4A1**, GAB1, VEGF-A)* and nitric oxide regulation (*THBS1**, VEGF-A*). We also observed that M2c marker genes, such as *THBS1* and *FPR1*, were upregulated compared to naïve M2 macrophages. Furthermore, since NAFLD has been associated with lipid metabolism, we tried to focus on those specific biological processes that were regulated in MERTK^+/hi^ M2c macrophages. It is worth noting that the cellular lipid catabolic process ranked as the 13th was enhanced in MERTK^+/hi^ M2c macrophages, which could have a beneficial effect on NAFLD. Thus, the transcriptomic profile of baicalin-induced MERTK^+/hi^ M2c macrophages is fundamentally distinct to that of the original M2 macrophage subset, and the major changes were a reduction in cell cycles, and an enhancement in the cellular response to vascular endothelial growth factor.

## 3. Discussion

Despite the enormous therapeutic strategies available to combat NAFLD, one in four people worldwide were still living with it [42]. With no approved treatments for this pathology, NAFLD remains persistent and difficult to treat. Its severe form, NASH, is one of the leading causes of hepatic transplantation because of its latent progression towards becoming cirrhosis and hepatocellular carcinoma [43]. The role of immune cells in response to liver injury is undeniable [44,45]. As part of the innate immune system, and acting as the first line of defense, macrophages have been shown to play an essential role in the injury and repair mechanisms during liver damage. During NAFLD, injury-induced modulations of the microenvironment in liver tissue, such as soluble mediators released by activated/stress cells and the accumulation of dead cells, influence the phenotypes of macrophages and determine their involvement in the aggravation of liver injury or the restoration of liver functions [46]. To date, M2 macrophages have been known to exhibit restorative and regenerative capabilities and, therefore, their potential application as a cellular therapy is very promising [14,47,48]. In this study, we demonstrated that baicalin promotes M2-like macrophages into specific M2c subset macrophages with high MERTK expression. The injection of baicalin-induced MERTK^+/hi^ M2c macrophages alleviates NAFLD through a possible mechanism involving the upregulation of serum HDL and the downregulation of NAFLD-associated genes in the liver and, therefore, decreases overall proinflammatory *TNFα*, the pro-fibrotic genes *COL1A1* and *FN*, as well as the lipid regulator *PPAR**ɣ* (as illustrated in Figure 7).

The prevalence of NAFLD has been associated with obesity, type 2 diabetes, and even with cardiovascular disease. The accumulation of fat and inflammation in the liver is the hallmark of NAFLD. In the present study, we have demonstrated excessive weight gain in mice fed an atherogenic diet. Because of the relatively low intensity of the treatment administered by injection of the MERTK^+/hi^ M2c macrophages (one dose per week from week four to week six), the amount of food intake for appetite among treated and untreated groups was presumed to be indistinguishable. Alternatively, we recorded mice welfare every two days after MERTK^+/hi^ M2c macrophage administration. The mice had remained active with no significant signs of stress or pain, such as squinted eyes, pulled-back ears, nose or cheek bulges, or whisker changes [49]. On account of the consistency in food consumption and the unchanged activity, we deduced that MERTK^+/hi^ M2c macrophages contributed to the weight reduction in mice in treatment. Consistent with our study, Matsuzawa et al. (2007) also revealed that an atherogenic diet fed to mice could induce fat accumulation, dyslipidemia, lipid peroxidation, and cellular ballooning in the liver, which upregulates the hepatic expression of genes for fatty acid synthesis, oxidative stress, and inflammation [5]. Moreover, several studies also reveal that an atherogenic diet is capable of inducing NAFLD in the mouse model because of the high percentage of cholesterol [30,31,50,51]. It has been shown that NAFLD is characterized by increased serum triglycerides, dense low-density lipoprotein (LDL nontype A) particles, and low high-density lipoprotein (HDL) cholesterol [52]. Our findings also revealed dyslipidemia in mice fed an atherogenic diet that was characterized by the increase of serum triglycerides and low-density lipoproteins (vLDL and LDL). However, the injection of MERTK^hi/+^ M2c macrophages could further increase the level of serum high-density lipoprotein (HDL), which was not shown in the normal diet and atherogenic diet groups. HDL, a negative predictor of NAFLD, has been known to offer hepatoprotective properties and prevent other NAFLD-related diseases, such as atherosclerotic cardiovascular disease [53]. As MERTK^hi/+^ M2c macrophages secreted anti-inflammatory molecules and pro-resolving lipid mediators (LM), M2 macrophages may enhance HDL production in the liver. Serum HDL mediates reverse cholesterol transport, and it is known to be protective against atherosclerosis. In addition, HDL has potent anti-inflammatory properties that may be critical for protection against other inflammatory diseases [54]. It has been shown that HDL suppressed the polarization of macrophages to an M1 phenotype, as indicated by a decrease in the expression of the M1-specific cell surface markers, CD64 and CD192, as well as the M1-associated inflammatory genes *IL-6*, *TNFα*, and *MCP-1* (*CCL2*). HDL also suppressed M1 function by reducing ROS production [55].

Because of the high incidence of NAFLD worldwide, liver biopsy is clearly impractical for routine diagnosis and staging. However, liver biopsy remains the “gold standard” for making a definitive diagnosis in the clinical trial setting and for research purposes [56,57]. With regard to scoring the severity of a liver biopsy to diagnose NAFLD, Kleiner et al. (2005) have suggested that the initial grading for NAFLD activity should start from mild, moderate, and severe, as the bases of the three characteristics of NAFLD (steatosis, hepatocytes ballooning, and inflammation) (Table 1) [39]. Hepatocellular steatosis is classified into two forms: the macrovesicular and the microvesicular. The former is identified by a single large fat droplet that occupies the cytoplasm of hepatocytes, pushing the nucleus to the periphery, while the latter has the cytoplasm of hepatocytes filled with tiny lipid droplets and the nucleus is located centrally in the cells. Hepatocyte ballooning is also common in NAFLD, characterized by swollen hepatocytes and wispy cytoplasm. Following steatosis and hepatocyte ballooning, an excess mixed population of inflammation cells may appear near the portal and lobular areas, exacerbating the disease [58]. Our results demonstrated that the atherogenic-diet group showed steatosis, ballooning degeneration, severe inflammation, and foci formation compared to the normal diet group, whereas treatment with MERTK^+/hi^ M2c macrophages alleviated the severity of NAFLD, especially the inflammation and foci formation, and decreased the score of NAFLD activity compared to the atherogenic-diet group. Additionally, we also found that the expression of *COL1A1*, which has been widely considered as one of the candidate genes for NAFLD [59], was also downregulated in the treatment group compared with the atherogenic-diet group. Fibronectin (*FN*), a glycoprotein associated with the extracellular matrix (ECM), was also downregulated significantly compared to that without treatment. The presence of fibronectin itself may be used to predict the progression of fibrosis at the early stages in the obese patient with a NASH record [60]. Furthermore, *PPARɣ**,* a key regulator for lipid metabolic and adipogenic factors, was also downregulated in the treatment group. Similar to our results, studies with mouse models with obesity and diabetes developing fatty livers are associated with a higher expression of *PPARɣ* and fat accumulation in the liver [61,62,63]. Additionally, the proinflammatory gene, *TNF**α*, was also suppressed following the treatment with MERTK^+/hi^ M2 macrophages. TNFα plays a major role in the development of NAFLD progression by upregulating key molecules related to lipid metabolism, inflammatory cytokines, and fibrosis in the liver [64]. Hence, our baicalin-induced MERTK^+/hi^ M2c macrophages may have hepatoprotective properties backwards from fatty liver disease. Furthermore, a study by Bai and colleagues (2017) has shown that M2-like macrophages protect hepatocytes against TNFα/D-GaIN-induced apoptosis in vitro, and confer hepatoprotection against the acute challenge of CCL4 [65]. Additionally, an in vivo study using a mouse fed a high-fat diet also demonstrated that M2 macrophages may promote M1 macrophage death. In this case, M2 macrophages secreted IL-10 to promote selective M1 death through a mechanism involving the activation of arginase in highly inducible nitric oxide-synthase-expressing M1 macrophages [66]. Although a lot of studies have been conducted to elucidate the mechanism of pro-resolving M2 macrophages, further research as to how the gene and the protein in M2 macrophages directly or indirectly contributes to the regulation of tissue repair and wound healing is needed.

As we mentioned before, MERTK^+/hi^ M2c macrophages could alleviate NAFLD progression. However, the molecular mechanisms behind it remains unknown. Transcriptomic analysis of MNC-derived BMDMs with baicalin induction indicated that MERTK^+/hi^ M2c macrophages have a lower expression of the cell-cycle-associated gene, suggesting a lower proliferative state compared to its original M2 macrophages. Meanwhile, baicalin also enhanced the cellular response towards vascular endothelial growth factor stimulus, which may contribute to macrophage recruitment, M2 polarization, and enhance angiogenesis, which promotes wound healing [67,68]. Moreover, the cellular lipid catabolic process was also enhanced in MERTK^+/hi^ M2c macrophages, suggesting the role of efferocytosis. Efferocytosis results in a significant accumulation of lipid inside the macrophage, yet the macrophage continues to function. This suggests that during efferocytosis, macrophages have pathways to ameliorate the high lipid load [69]. Further, transcriptomic analysis data suggests that some of the genes might play significant roles in alleviating the NAFLD burden. Moreover, baicalin-induced MERTK^+/hi^ M2c macrophages may secrete protein and affect neighboring cells directly or indirectly. For instance, *SERPINE1*, a plasminogen activator inhibitor-1 that serves as a critical downstream target of p53 for the replicative senescence (GO:0090399) via the inhibition of PI(3)K-PKB-GSK3beta and the cyclin D1 pathway was downregulated in our study [70]. NAFLD has been associated with metabolic syndrome, which provokes premature hepatocytes senescence, worsening the disease [71]. Consistent with the results, the inhibition of SERPINE1 partially alleviated NAFLD by preventing steatosis events in mice fed with a high-fat, high-cholesterol, high-sugar (HFHS) diet, or a methionine-and choline-deficient (MCD) diet [72]. Additionally, MERTK^+/hi^ M2c macrophages may contribute to a decrease in immune response, such as inflammation, which is the hallmark of NAFLD. For instance, ∆6-fatty acid desaturase 2 (*FADS2*), which regulates the polyunsaturated fatty acid (PUFA) synthesis and its downstream, eicosanoid, were downregulated in MERTK^+/hi^ M2c macrophages. FADS2 has been shown to play an essential role in inflammation response. A study from Stoffel et al. (2008) demonstrates that, at the site of inflammation, *FADS2*^−/−^ mice failed to transform arachidonic acid into leukotriene B4 (LTB4), a potent chemokine promoting the migration of innate immune cells into inflamed tissues [73]. Additionally, FADS2 has a strong correlation with prostaglandins, such as prostaglandin E2 (PGE2), which are generated by the action of cyclooxygenase (COX) isoenzyme during inflammation [74]. Therefore, the downregulation of *FADS2* expression level may have a beneficial role to play in reducing inflammatory mediators, although further studies are needed to elucidate the exact impact and roles of FADS, especially FADS2, in M2 macrophages. Furthermore, C-C motif chemokine receptor 5 (*CCR5*), which has been shown to mediate leukocyte infiltration and determine inflammatory response, was also downregulated. A study conducted by Kitade et al. (2012) reveals that CCR5 and its ligands are upregulated in the white adipose tissue (WAT) of genetically (*ob/ob*) and HF-diet-induced obese (DIO) mice, worsening the inflammation [75]. On the contrary, *CCR5^−/−^* mice are protected from insulin resistance, hepatic steatosis, and diabetes induced by HF feeding. Moreover, the inflammatory signals in the WAT of *CCR5^−/−^* mice were attenuated. The phosphorylation of Jun NH_2_-terminal kinase (JNK) and p38 mitogen-activated protein kinase (MAPK) were downregulated in *CCR5^−/−^* mice on both normal chow (NC) and HF diets. Nuclear factor-κB (NF-κB) p65 phosphorylation was also suppressed in *CCR5^−/−^* mice on an HF diet. Strikingly, flow cytometry analysis also revealed an increase in CD11c*^−^* MGL1*^+^* M2-type ATMs in obese *CCR5**^−/−^* mice, suggesting that the depletion of *CCR5* causes a shift to an M2-dominant phenotype [75].

MERTK signaling has been considered essential in suppressing innate immune responses. In mice with a bleomycin-induced lung injury, the reduced expression of membrane-bound MERTK leads to the activation of inflammatory responses and the insignificant recruitment of inflammatory cells, mediated by the release of soluble TNFα and IL-1 [76]. Additionally, MERTK expression has been associated with the ability of macrophages to perform efferocytosis and phagocytosis that could lead to decreases in inflammation and disease progression [14]. A recent study has also elucidated a strong correlation between high MERTK expression and cholesterol efflux. In this way, a combination of aminolevulinic acid (ALA) and sonodynamic therapy (SDT) could increase the efferocytosis capacity of macrophages toward the foam cell, which further provides natural ligands for PPARγ and stimulates cholesterol efflux out of phagocytes via the PPARγ-LXRα-ABCA1/ABCG1 pathway [29]. In NAFLD, cholesterol efflux capacity (CEC) is impaired, followed by an elevation of apoB lipoprotein and a low HDL, which could worsen the disease progression [77]. In this context, treatment with MERTK^+/hi^ M2c macrophages might be a promising candidate for resolving these particular problems. Despite the importance of these receptors, the presence of MERTK itself in the liver may exhibit a poorer prognosis for NAFLD patients. A recent study has proven that the presence of high MERTK, particularly in the Kuppfer cells, may increase the prevalence of fibrosis by the activation of hepatic stellate cells (HSCs) via the ERK-TGFβ pathway [78]. On the contrary, our results demonstrated that baicalin-induced MERTK^+/hi^ M2c macrophages, as an exogenous source, could alleviate NAFLD. In this context, an atherogenic diet with high cholesterol and cholate content was used to induce NAFLD. While a lot of studies have been exclusively focused on a high-fat, or Western, diet [71,75,78], atherogenic-diet-induced NAFLD, and its difference from HFD, remain elusive. Moreover, our study was focused on baicalin-induced M2 macrophages, which have been shown to possess a beneficial effect compared to the endogenous M1-like Kupffer cells. It has been shown that during diet-induced liver injury, tissue-resident macrophages exhibit a classically activated M1 phenotype predominate, which exacerbates liver injury [79]. Nonetheless, this study provides an insight into MERTK^+/hi^ M2c macrophages as a cell-based therapy that could alleviate NAFLD rather than worsen it.

## 4. Materials and Methods

### 4.1. NAFLD Mouse Model

Seven-week to eight-week-old female C57BL/6 mice were purchased from the National Laboratory Animal Center (Taipei, Taiwan). All of the experimental mice procedures were approved by the Institutional Animal Care and Use Committee at the National Pingtung University of Science and Technology and housed in a clean conventional animal facility at 22 °C with a 12 h light/ dark cycle. The mice were randomly divided into a normal chow diet group, an atherogenic diet group (TestDiets, St. Louis, MO, USA), and an atherogenic diet + MERTK^+/hi^ M2c macrophages group. While the treatment group received retro-orbital injections of 100 μL 10^6^ MERTK^+/hi^ M2c macrophages, the atherogenic diet group received sham retro-orbital injections of 100 μL PBS by the same procedure as the treatment group at weeks 4, 5 and 6 (total three injection times). The mice in each group were sacrificed in the seventh week after being fed an atherogenic diet.

### 4.2. Preparation of MERTK^+/hi^ M2c Macrophage

The bone marrow was flushed from the tibias and femurs of 6–8-week-old C57BL/6J mice with phosphate-buffered saline (PBS) containing BSA 0.5%. The mononuclear cell of the peripheral blood and bone marrow were isolated by Ficoll-Hypaque (GE Healthcare, Uppsala, Sweden) according to the manufacturer’s instructions. Briefly, the peripheral blood or bone marrow cells were layered with 3 mL Ficoll-Hypaque in a 15 mL tube and centrifuged with 400× *g* for 40 min at 18 °C. The interphase layer containing mononuclear cells was harvested and washed with PBS, then the cells were centrifuged with 100× *g* for 10 min at 18 °C (twice). The mononuclear cells were plated in a dish culture with RPMI 1640 (Corning, Manassas, VA, USA) plus 10% FBS (fetal bovine serum) supplemented with 50 ng/mL recombinant mouse M-CSF (315-02 PeproTech, Rocky Hill, NJ, USA) at 37 °C and 5% CO_2_ for 7 days. After 7 days of treatment by M-CSF, bone-marrow-derived-macrophage (BMDM) cells were plated in RPMI 1640 plus 10% FBS and incubated overnight at 37 °C and 5% CO_2_ (5 × 10^5^/well, 6-well plate). After incubating overnight, BMDM cells were treated with Baicalin (50 μM) for 24 h at 37 °C and 5% CO_2_. After incubation, cells were washed with PBS and collected for surface protein expression analysis by flow cytometry. (BD FACSAria II, BD Biosciences, San Jose, CA, USA) to characterize the M2 macrophage phenotype, especially with the higher expression of MERTK^+/hi^ and with the expression level of CD11b (Biolegend, San Diego, CA, USA).

### 4.3. Surface Protein Expression of MERTK^+/hi^ M2c Macrophage

Harvest cells (~1 × 10^6^cells) were washed twice with PBS (3 mL/tube), centrifuged at 300× *g* at 4 °C for 5 min, and then the supernatant was removed. Various anti-mouse mAb, conjugated with FITC-anti-CD11b (1:100), and PE-anti-MERTK (1:50), (Biolegend, San Diego, CA, USA) was added and light was avoided. The cells were stained for 30 min on ice with added antibodies. Flow cytometry was performed using FACS BD II ARIA, and data were analyzed with FlowJo software V.10.1 (Tree Star, Ashland, OR, USA).

### 4.4. Plasma Lipid Measurements

Mice were fasted overnight before collecting the blood through retro-orbital veins under isoflurane anesthesia. Blood was collected with a capillary tube containing heparin, or EDTA, as an anticoagulant agent. Then the blood was centrifuged for 30 min at 2–8 °C at 3000 rpm. To eliminate the platelet effect, further centrifugation for 10 min at 2–8 °C at 10000× *g* was conducted. The supernatant was collected and subjected to the HDL and LDL/VLDL Cholesterol Assay Kit (Cell Biolabs, San Diego, CA, USA) measurement by enzymatic assays following the manufacturer’s protocol. The optical density (OD) was measured with a microplate reader (Biochrom, UK).

### 4.5. Histological Analysis of NAFLD

Mice in all groups were sacrificed after seven weeks of feeding. The livers of all the mice were collected and examined. The tissues were embedded in an optimal cutting temperature compound. Briefly, for paraffin-embedded tissue, the samples were perfused with saline followed by 10% formalin overnight in a cassette. The samples were washed with ddH_2_O for 4 h and immersed in the respective fixative (70%, 80%, 90%, 95%, 100% ethanol) for 1 h (twice). Next, the sample was placed in xylene for 1 h (three times). After that, the samples were placed in the paraffin for about 3 h and the samples were embedded in a cassette containing paraffin. For histology analysis, the paraffin-embedded tissue was cut at a 5–10 μm thickness. For each liver, several sections were discarded from the initial cutting and approximately 3–5 samples were collected on the slides. All the sections were subjected to modified hematoxylin and eosin staining (Sigma Aldrich, Steinheim, Germany), as well as modified Masson’s trichrome staining (Sigma Aldrich, MO, USA) following the manufacturerer’s protocols.

### 4.6. NAFLD Activity Score

Liver structure evaluation was determined by an inverted microscope Olympus IX71 (Tokyo, Japan) to evaluate the grade of the induced histopathological changes using a histological scoring system for NAFLD [40]. Briefly, three to five different sections per liver in each group were evaluated and graded with the NAFLD scoring system. This scoring system consists of three major characteristics in the NAFLD liver: hepatic steatosis (S), range from 0–3; inflammation (L), range from 0–3; and the ballooning of hepatocytes (B), range from 0–2. Livers with a <5% steatosis area, no inflammation, and ballooning degeneration, were regarded as 0 scores. Livers with a 5–33% steatosis area, with <2 foci of inflammation/20× field, and few ballooning degenerations are graded as 1. Livers with a >33%–66% steatosis area, with a 2–4 inflammation foci/20× field, with a lot of ballooning degeneration, were scored as 2. Livers with more than 66% of steatosis area and with more than four inflammation foci were scored as 3. The total scores indicate the sum of all the hepatic changes, according to the following equation [Total NAFLD Activity Score (NAS) = S + L + B], and were graded as severe (≥5), moderate (3–4), and mild (<3) [39]. To confirm the severity of liver damage, fibrosis and necrosis evaluations were also conducted separately with the range from 0–4. Histological findings, such as mild pericellular fibrosis, moderate pericellular fibrosis, portal/periportal fibrosis without pericellular fibrosis, pericellular and portal fibrosis, bridging necrosis, and cirrhosis are graded as 1a, 1b, 1c, 2, 3 and 4, respectively [40].

### 4.7. RNA Extraction and Quantitative Real-Time Polymerase Chain Reaction

The total RNA of the liver samples was prepared using Trizol reagent (Invitrogen, Waltham, MA, USA) according to the manufacturer’s protocol. Total RNA (1 μg) was used in the reverse transcription (RT) reaction by the iScript cDNA Synthesis Kit (Bio-Rad, Hercules, CA, USA). The quantitative real-time PCR was performed using the KAPA SYBR^®^ FAST qPCR Master Mix (2X) Kit (KAPA Biosystem, Wilmington, DE, USA) according to the manufacturer’s protocol. Quantitative PCR reactions were performed using a QIAGEN Rotor Gene Q Real-Time PCR. The primer sequences (5′–3′; forward, reverse) are shown in Table A1 (Appendix A). The relative gene expression of target genes was normalized to β-actin (the housekeeping gene). The constitutive expression of the β-actin gene level was not altered in relation to mice fed a normal diet [80]. The data were calculated using the standardized mRNA level comparative method 2^−ΔΔCt^.

### 4.8. Transcriptome Profiling

Total RNA was isolated from cultured macrophages using TriPure Isolation Reagent (Roche) according to the manufacturer’s instructions. Library preparation was carried out following the TruSeq Stranded mRNA Library Prep Kit (Illumina, San Diego, CA, USA) according to the manufacturer’s protocol. The quality of the libraries was sized and checked for adapter contamination using the Agilent Bioanalyzer 2100 system and a Real-Time PCR system. These libraries were then sequenced using the Illumina NovaSeq 6000 platform with 150 bp paired-end reads generated by Genomics, BioSci & Tech Co., New Taipei City, Taiwan. The raw sequencing reads were then sorted using the Trimmomatic program (version 0.39). After sorting, the remaining reads are called “clean reads”. The clean reads are mapped to reference using Bowtie2 tools (version 2.3.4.1). The gene’s expression level is quantified by a software package called RNA Sequencing by Expectation-Maximization (RSEM) (version 1.2.28), and differentially expressed genes (DEGs) were identified by EBSeq (version 1.16.0) software at a target false discovery rate (FDR) of 0.05. The functional enrichment analysis of the gene ontology (GO) terms among gene clusters was implemented in an R package called clusterProfiler (version 3.6.0), and Enrichr [38] databases were used to determine the enrichment of biological processes, on the basis of the genes that were significantly upregulated or downregulated in the MERTK^+/hi^ M2c population in response to treatment with baicalin. The datasets presented in this study can be found in online repositories (NCBI BioProject PRJNA736111: https://dataview.ncbi.nlm.nih.gov/object/PRJNA736111?reviewer=mu616g47t95eapje8ers19ggr5, accessed on 16 June 2021).

### 4.9. Statistical Analysis

All results were presented as mean ± SEM. Comparison among groups was made using one-way ANOVA test analysis. Statistical analysis was performed with GraphPad Prism 7 (GraphPad Software, Inc. La Jolla, CA, USA). *p*-values less than 0.05 were considered statistically significant.

## 5. Conclusions

In summary, we here provide evidence that MERTK^+/hi^ M2c macrophages exhibit a therapeutic role for NAFLD. MERTK^+/hi^ M2c macrophage treatment may contribute to enhanced HDL production in the liver and the modulation of the circulating T cells. Treatment also decreased the overall NAS score, as well as the inflammatory events and hepatic steatosis. Treatment with MERTK^+/hi^ M2c macrophages also suppressed the onset of inflammation by reducing proinflammatory genes, such as TNFα, lipid regulator, and adipogenic factor PPARɣ, and the profibrotic FN and COL1A1in the liver. Transcriptomic analysis revealed that baicalin-induced MERTK^+/hi^ M2c macrophages have distinct features compared to the naïve M2 macrophages, in a reduced cell cycles profile, an enhanced cellular response to the VEGF, and a cellular lipid catabolic process. Various genes related to lipid, fat, and metabolic activity, such as SERPINE1 and FADS2, might participate in the recovery process of NAFLD. Finally, the downregulation of cytokines and inflammation-associated genes, such as CCR5, may promote a pro-resolving milieu in the NAFLD liver. Altogether, MERTK^+/hi^ M2c macrophages could be a promising cell-based autologous therapy for NAFLD.

## Figures and Tables

**Figure 1 ijms-22-10604-f001:**
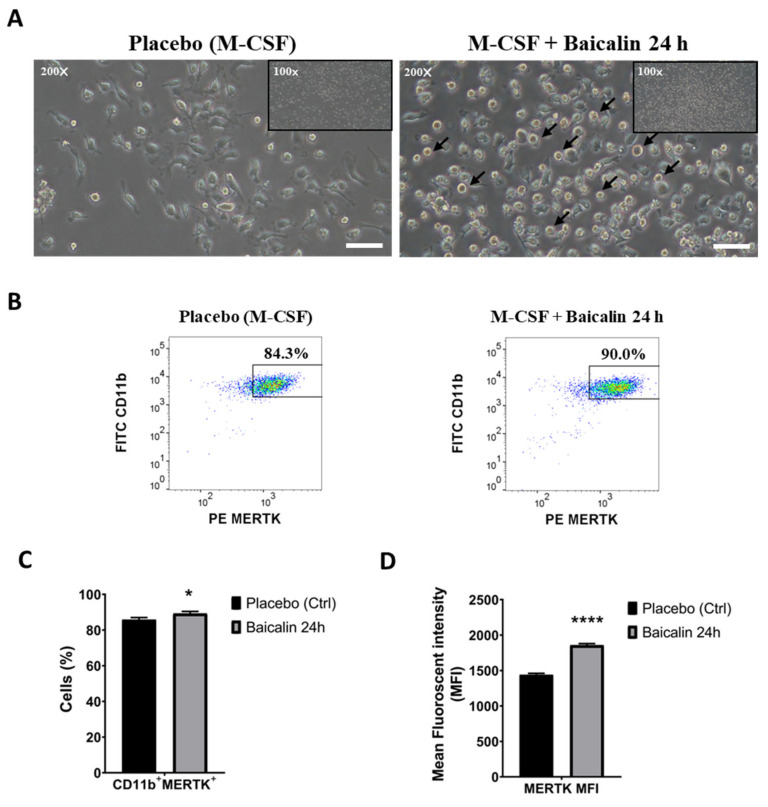
MERTK^+/hi^ M2c macrophages polarized by baicalin prior to injection into an NAFLD mouse model. (**A**) M2c macrophage polarization by baicalin shows a specific round shape. Magnification, 200× (large pictures), 100× (miniatures), scale bar: 50 μm (**B**,**C**). M2 macrophages from bone marrow were stained with FITC anti-mouse CD11b and PE anti-mouse MERTK. The surface markers (CD11b, MERTK) on M2 macrophages were determined by flow cytometry. (**D**) Flow cytometry analysis of mean fluorescence intensity (MFI) of MERTK. Data are expressed as mean ± SEM, *n* = 3. * *p* < 0.05 compared with the placebo group; **** *p* < 0.0001 compared with the placebo group.

**Figure 2 ijms-22-10604-f002:**
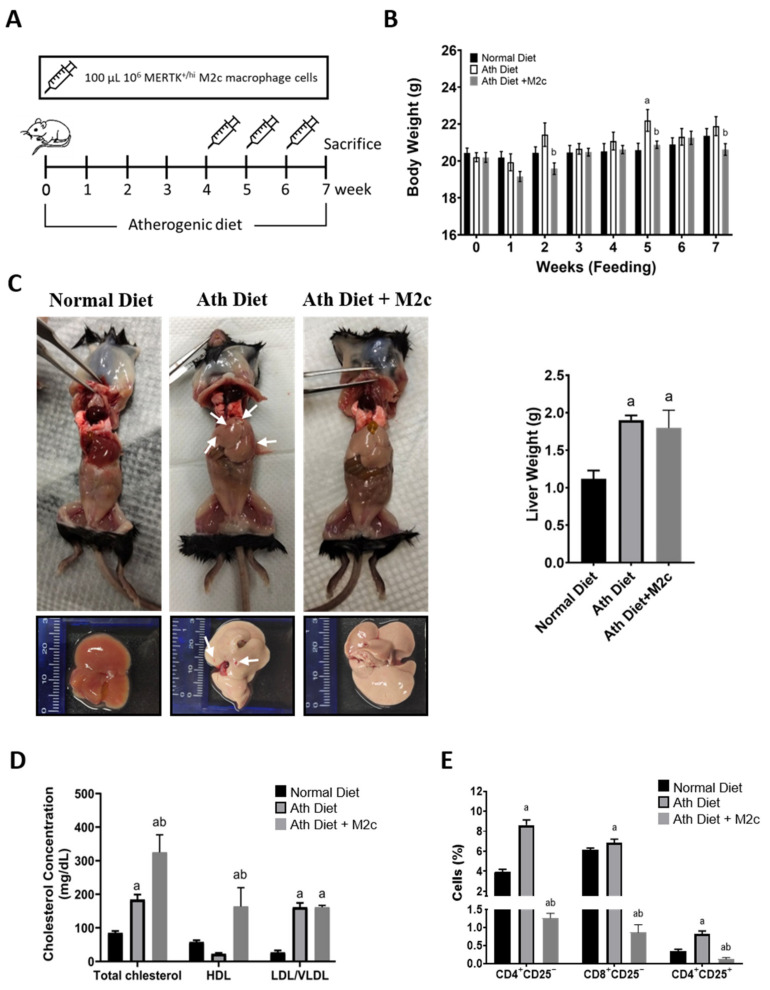
The C57BL/6 mice were fed with an atherogenic diet for seven weeks and were injected with MERTK^+/hi^ M2c macrophages for three weeks (one dose/week). (**A**) Treatment process illustration of NAFLD mouse model. The mice were fed an atherogenic diet throughout the study and were injected with 100 μL of 10^6^ MERTK^+/hi^ M2c macrophages for the last three weeks before being sacrificed; (**B**) The body weight of mice was measured every week after feeding; (**C**) The gross appearance of livers from representative mice in each group. The livers were weighed and compared with the normal chow diet group. Indication of hemorrhage with red spots and scarring were also observed (arrow). Note that the color may differ because of the camera processing/lighting; (**D**) Cholesterol, HDL, and LDL/vLDL levels in plasma. Plasma samples from mice fed normal and atherogenic (Ath) diets were manifested by ELISA; (**E**) Flow cytometry analysis of CD4^+^CD25^−^, CD8^+^CD25^−^, and CD4^+^CD25^+^ T cells in peripheral blood manifested by flow cytometry. Representative data were shown and determined by mean ± SEM, *n* = 10. ^a^ *p* < 0.05 compared with normal diet group; ^b^ *p* < 0.05 compared with Ath diet group.

**Figure 3 ijms-22-10604-f003:**
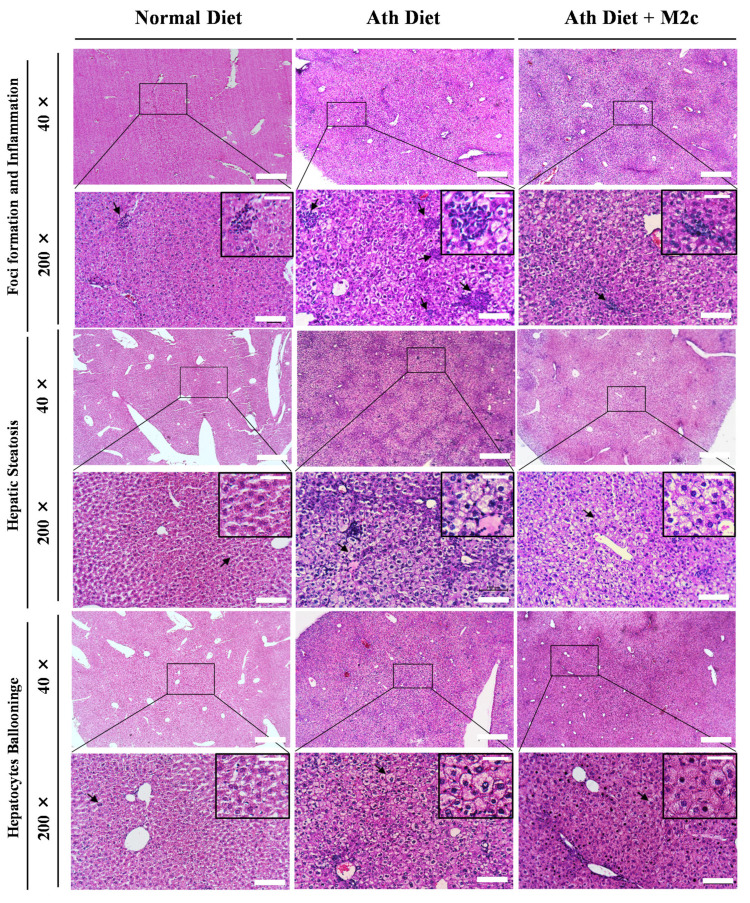
Representative histopathological analysis of NAFLD liver biopsy. The atherogenic diet group showed various indications of liver damage/injury and NAFLD compared to the treatment group, which received a weekly dose of MERTK^+/hi^ M2c macrophages. Compared to the treatment group, the atherogenic diet group demonstrated severe inflammation with a lot of clusters of heterogeneous inflammation cells, which usually consist of lymphocytes, neutrophils, eosinophils, and macrophages. However, hepatocyte ballooning was still prominent in both groups. Indications, such as foci formation and inflammation, hepatic steatosis, and hepatocyte ballooning were used to determine NAFLD severity by the NAS scoring system. Scale bar: 500 μm, magnification 40×; 100 μm, magnification 200×; 50 μm, detailed miniatures.

**Figure 4 ijms-22-10604-f004:**
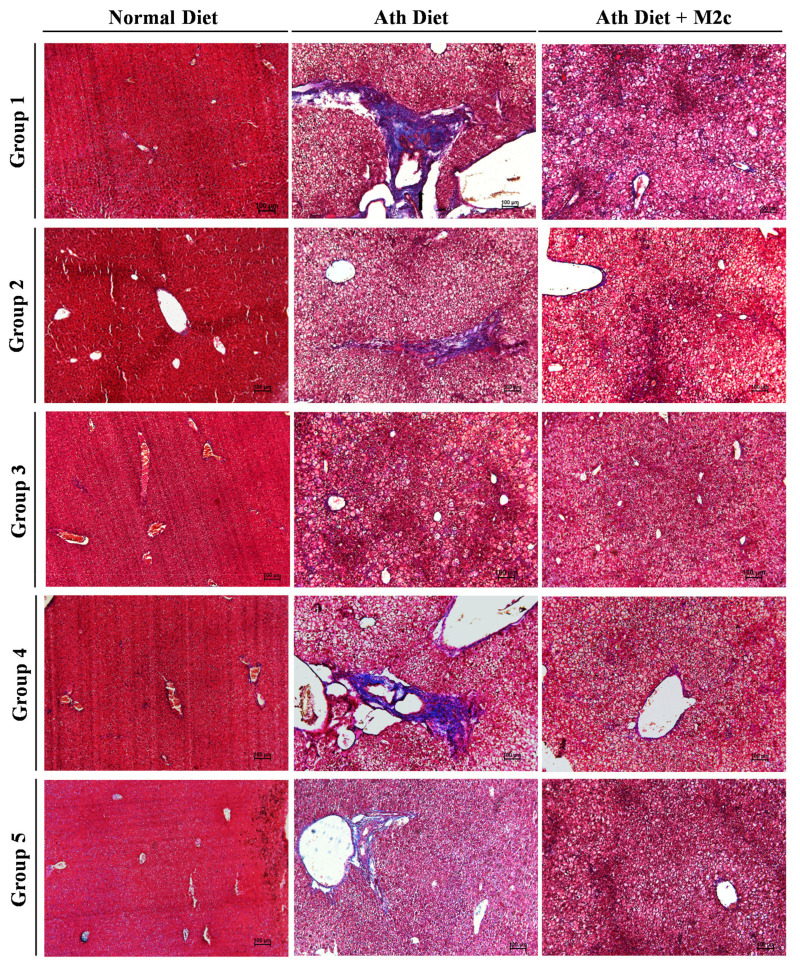
Representative histopathological analysis of fibrosis in a NAFLD liver. Modified Masson’s trichrome staining was used to stain the connective tissues and collagen (blue color). The majority of the atherogenic diet group demonstrated both pericellular and portal/periportal fibrosis marked by the collagen deposition around the vessel and sinusoidal area. Note that treatment with MERTK^+/hi^ M2c macrophages could reduce the collagen development around the vessel and sinusoidal area. Histological findings, such as mild pericellular fibrosis, moderate pericellular fibrosis, portal/periportal fibrosis without pericellular fibrosis, pericellular and portal fibrosis, bridging necrosis, and cirrhosis are graded as 1a, 1b, 1c, 2, 3, and 4, respectively. Scale bar: 100 μm, magnification 100×.

**Figure 5 ijms-22-10604-f005:**
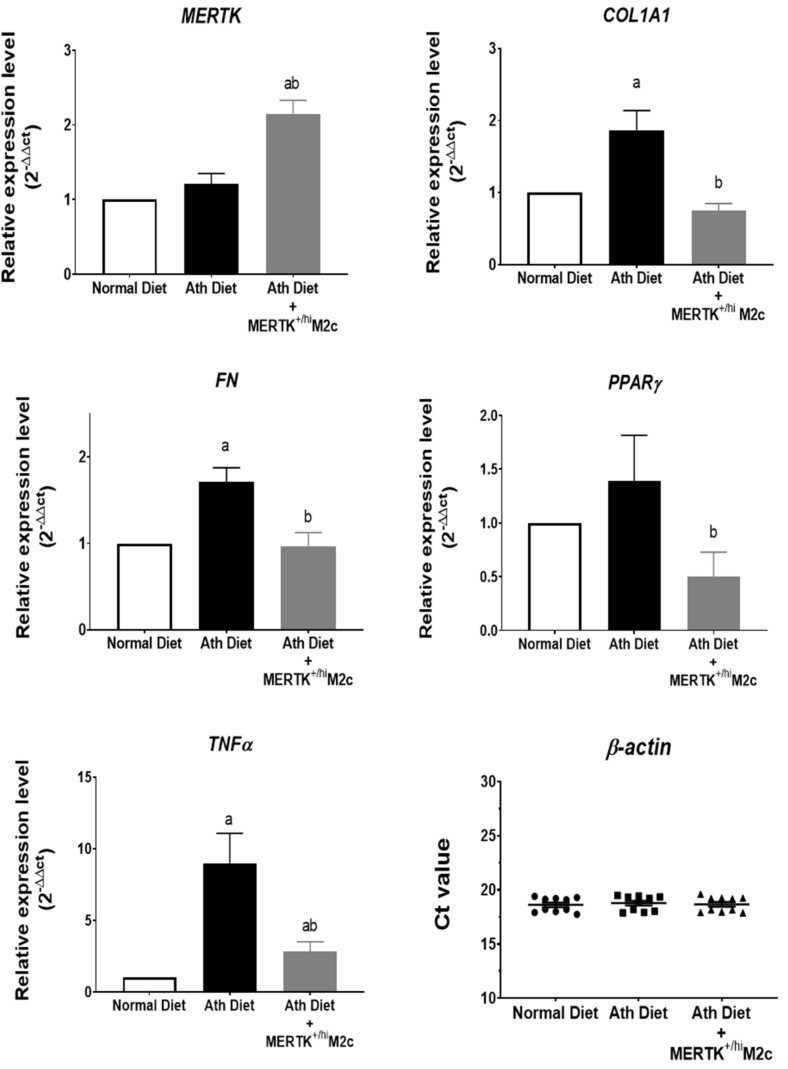
MERTK^+/hi^ M2c macrophages suppress the profibrotic activity in a NAFLD liver. RT-PCR was utilized to evaluate MERTK receptor-, pro-fibrotic-, lipid metabolism-, and inflammatory-associated gene expression, and then normalized to that of *β-actin*. The relative gene expression level of *MERTK* was higher in the treatment group. The transcript level of profibrotic collagen type 1 alpha (*COL1A1*), fibronectin (*FN*), and *PPARɣ* were downregulated in mice treated with MERTK^+/hi^ M2c macrophages. A significant decrease in tumor necrosis factor-alpha (*TNFα*) was also observed in the treatment group compared to that of the atherogenic diet group. (*n* = 5 for each group). ^a^ *p* < 0.05 compared with the normal diet group; ^b^ *p* < 0.05 compared with the Ath diet group. Data are presented as mean ± SEM.

**Figure 6 ijms-22-10604-f006:**
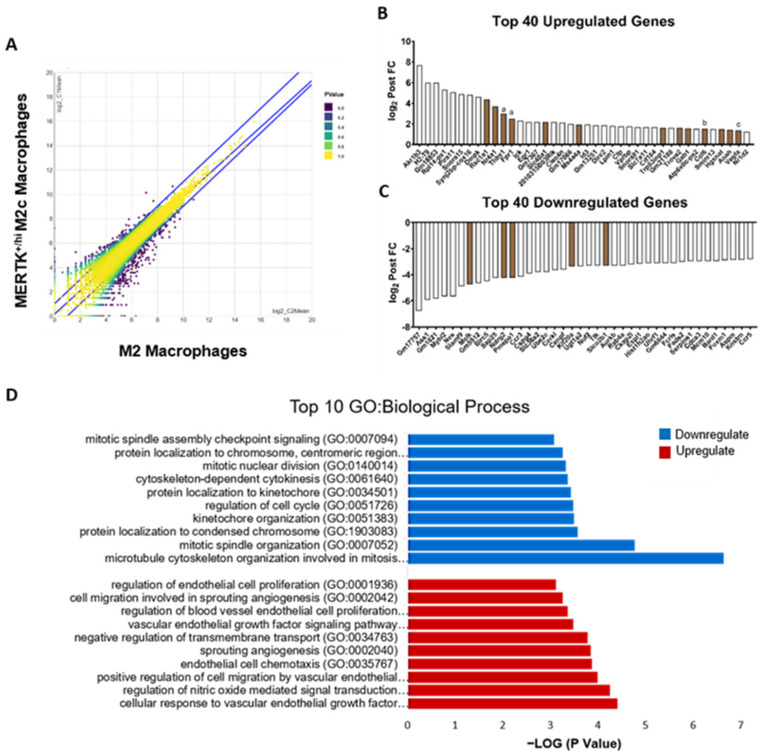
Transcriptome analysis of M-CSF + baicalin-induced MERTK^+/hi^ M2c vs M-CSF-induced M2 macrophages. **(A**) Venn diagram of differentially expressed genes in both MERTK^+/hi^ M2c macrophages and M2 macrophages. (**B**,**C**) Histograms of the top 40 upregulated and downregulated differentially expressed genes (DEGs). Note that the brown mark represents the M2-associated gene. The labels of A, B and C correspond to the M2c, M2a, and M2d genes, respectively. (**D**) Top 10 upregulated and downregulated GO: Biological process when the original M2 macrophages were compared with the MERTK^+/hi^ M2c macrophages. GO ontology analysis and enrichment was determined by Enrichr software. False discovery rate [FDR] ≤ 0.05.

**Figure 7 ijms-22-10604-f007:**
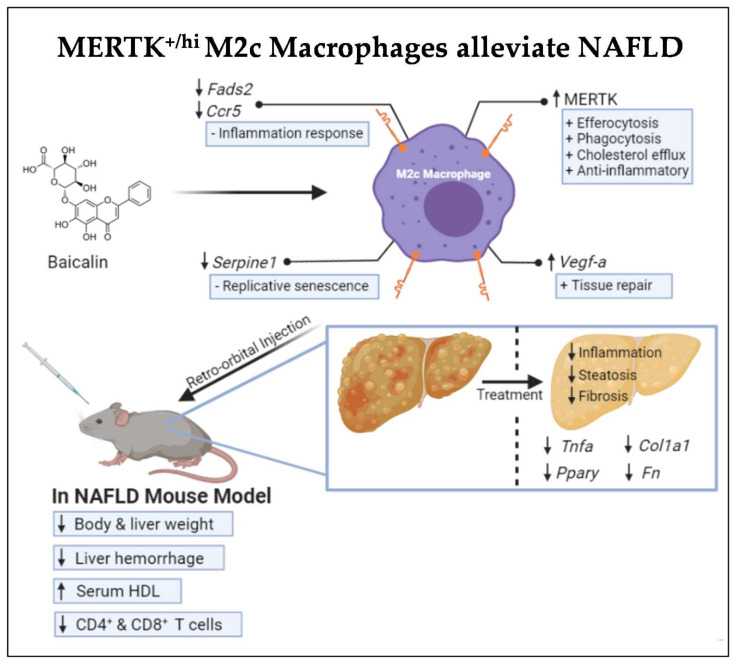
Graphical illustration of the factors that are involved in how MERTK^+/hi^ M2c macrophages alleviate NAFLD. Baicalin-induced MERTK^+/hi^ M2c macrophages have been strongly linked to enhanced efferocytosis, phagocytosis, cholesterol efflux, and anti-inflammatory properties. In addition, *VEGF-A*, a gene linked to the reparative response, was upregulated, while the inflammation-response-related genes (e.g., *FADS2* and *CCR5*), as well as the replicative senescence-associated gene (*SERPINE1*) were downregulated. Injection to the NAFLD mouse model has been shown to decrease the overall body and liver weight, increasing serum HDL, and is capable of exerting its T cell immunomodulation. Hence, the liver showed less inflammation, steatosis, and fibrosis histologically, which were proven by the reduction of NAFLD-associated gene expression (e.g., *TNFα*, *PPARɣ*, *COL1A1*, and *FN*) in the molecular events.

**Table 1 ijms-22-10604-t001:** NAFLD activity score (NAS) components of the treatment groups.

Score Components	Groups (Mean ± SEM)		
	Control Group	Atherogenic Diet Group	Atherogenic Diet + MERTK^+/hi^ M2c Macrophages
Steatosis			
Score	0.00 ± 0.00	1.86 ± 0.08 ^a^	0.96 ± 0.13 ^ab^
Extent	<5%	>5–33%	>5–33%
Inflammation			
Score	0.20 ± 0.20	2.66 ± 0.38 ^a^	1.16 ± 0.24 ^ab^
Extent	None	>4 foci/200×	<2 foci/200×
Ballooning degeneration			
Score	0.00 ± 0.00	2.00 ± 0.00 ^a^	1.26 ± 0.12 ^a^
Extent	None	Many	Many
Total Scores	0.20 ± 0.20	6.52 ± 0.39 ^a^	3.39 ± 0.37 ^ab^
**Fibrosis Stage**			
Necrosis and Fibrosis			
Score	0	2 ^a^	1b ^a^
Extent	None	Pericellular and portal/periportal fibrosis	Moderate pericellular fibrosis

NAS: NAFLD activity scores, *n* = 15 mice in total, *n* = 5 mice for each group, total score graded as severe (≥5), moderate (3–4), and mild (<3). NAFLD scoring and fibrosis stage were graded with the revised Brunt’s system [40]. All data are presented as means ± SEM (*n* = 5 mice/group). ^a^ *p* < 0.05 compared with normal diet group; ^b^ *p* < 0.05 compared with Ath diet group.

## Data Availability

Not applicable.

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
