# Peer review of "MERTK^+/hi^ M2c Macrophages Induced by Baicalin Alleviate Non-Alcoholic Fatty Liver Disease"

_ijms, 2021, doi:10.3390/ijms221910604_

Round 1

Reviewer 1 Report

The authors present a study of the effects of a novel cellular therapy, injection of baicalin-induced M2c macrophages, on the progression of NAFLD. Although a significant amount of work has been completed and the concept of the study is interesting, there are significant problems with experimental design and data presentation.  The data presented do not support the authors’ conclusions, and several major concerns need to be addressed.

1) There was a significant decrease in body weight in the M2c-treated group.  Was food consumption also decreased? This could account for differences in body weight gain in the M2c group, which could entirely explain the effect of the treatment on NAFLD progression. 

2) Did the control groups receive sham retro-orbital injections of the vehicle used for M2c injections? 

3) An important control would be injection of M2 compared to M2c macrophages ; i.e. does baicalin-induced M2c actually make a difference, or would M2 be sufficient?

4) It is not clear how the values for NAS and Fibrosis Stage are presented (Table 1).  Are the values averages of the tissue sections evaluated? If so, how many sections per mouse liver where scored, and what is the standard deviation or error range for each group?  What statistics were performed to determine whether differences between groups were significant?  Since the authors' conclusion that M2c treatment decreases NAFLD progression relies on these data, it is important that they are quantitative rather than qualitative.

5) Analyses presented in Table 1 are not consistent with gene expression data in Fig. 4 which show no induction of inflammation and fibrosis with atherogenic diet.  This discrepancy calls the validity of the mouse model into question and needs to be addressed.

6) The authors' conclusion regarding increased HDL as a mechanism for improvement in NAFLD progression is problematic. Wildtype C57Bl6 mice, like other rodents, and unlike humans, carry the majority of serum cholesterol in HDL. An increase in HDL in this model indicates an increase in serum cholesterol (also indicated by total the total serum cholesterol data in Fig. 2D), which is not an indicator of improved metabolic health. 

7) What was the sex of the mice used? Male?

8) Overall, the data presented do not support the working model in Fig.6.

Author Response

# ijms-1377125

The authors thank the reviewers for spending considerable amount of time and efforts on examining this manuscript and greatly appreciate the constructive comments and the suggestions made by the reviewers. The authors have tried to respond to all the comments and accommodate most of the suggestions in the revised manuscript.

The authors’ answer:

        The authors thank and appreciate the reviewer’s comments. Therefore, we’ve revised our manuscript according to reviewer advises.

Reviewer #1

Comments and Suggestions for Authors

The authors present a study of the effects of a novel cellular therapy, injection of baicalin-induced M2c macrophages, on the progression of NAFLD. Although a significant amount of work has been completed and the concept of the study is interesting, there are significant problems with experimental design and data presentation. The data presented do not support the authors’ conclusions, and several major concerns need to be addressed.

1) There was a significant decrease in body weight in the M2c-treated group. Was food consumption also decreased? This could account for differences in body weight gain in the M2c group, which could entirely explain the effect of the treatment on NAFLD progression.

ANS:

We would like to thank the reviewer for the feedback regarding the body weight of the mice. Indeed, we did not measure the food intake in each group and there was a fluctuation in the body weight of the mice in both treatment and untreated group. However, we observed that the food was reduced in every 2 days (we refill the food every 2 days for approximately 60 g). Besides, two days after injection, the mice remained active and did not show any significant sign of stress and pain such as squinted eyes, pulled-back ears, nose or cheek bulge, and whisker change. After injection, the treatment group did not show any significant reduction in body weight compared to the control group. However, compared with the M2c group, the atherogenic diet group showed a significant difference in body weight only at the fifth week and the seventh week, suggesting the treatment may somehow influence the body weight of each mice. We thank the reviewer's comment, we will measure the food intake for further study. The manuscript has been updated accordingly (Result section 2.2, Page 5)

2) Did the control groups receive sham retro-orbital injections of the vehicle used for M2c injections?  

ANS:

We really appreciate the comment given. As the sham group, we use PBS and injected it via retro-orbital injections to the atherogenic diet group. Indeed, we did not mention that atherogenic diet group received sham retro-orbital injections of PBS in the manuscript and therefore, we have made an update in the materials and methods as well as the result sections to clarify it. [page 14, line 573-574; page 4, line 152].

3) An important control would be injection of M2 compared to M2c macrophages; i.e. does baicalin-induced M2c actually make a difference, or would M2 be sufficient?

ANS:

We would like to thank the reviewer for the comment. M2 macrophages can be further subdivided into M2a (wound healing macrophages), M2b (protective and pathogenic roles, and secrete both pro- and anti-inflammatory cytokines), M2c (regulatory phenotype, repress inflammation and fibrosis, promote tissue repair), and M2d (mainly contributing to angiogenesis and metastasis, supporting tumor growth) subtype. In addition, M2c subtype has been known to express higher MERTK receptor which demonstrate efferocytosis and phagocytosis capability, and enhanced cholesterol efflux with anti-inflammatory activity [19, 22, 23]. Based on the facts that NAFLD has a strong correlation with excess inflammation response and impaired cholesterol efflux capacity (CEC), we first confirm the efficacy of MERTK+/hi M2c macrophages to alleviate NAFLD first, and therefore it is not our focus of study to compare the efficacy of M2 vs M2c macorphages. To our knowledge, this is the first study that offer the M2c macrophages cell-based therapy to alleviate such disease. We have made an update in the introduction section. [page 2, line 63; page 3; line124].

Rőszer, T. (2015). Understanding the mysterious M2 macrophage through activation markers and effector mechanisms. Mediators of inflammation, 2015.

Wang, L. X., Zhang, S. X., Wu, H. J., Rong, X. L., & Guo, J. (2019). M2b macrophage polarization and its roles in diseases. Journal of leukocyte biology, 106(2), 345-358.

Wang, Q., Ni, H., Lan, L., Wei, X., Xiang, R., & Wang, Y. (2010). Fra-1 protooncogene regulates IL-6 expression in macrophages and promotes the generation of M2d macrophages. Cell research, 20(6), 701-712.

4) It is not clear how the values for NAS and Fibrosis Stage are presented (Table 1).  Are the values averages of the tissue sections evaluated? If so, how many sections per mouse liver where scored, and what is the standard deviation or error range for each group?  What statistics were performed to determine whether differences between groups were significant? Since the authors' conclusion that M2c treatment decreases NAFLD progression relies on these data, it is important that they are quantitative rather than qualitative.

ANS:                     

We thank the reviewer for the feedback. The values averages of the tissue sections in each liver of every group were evaluated. In this case, approximately 3 sections per mouse liver in each group (n=15; n=5/group) were collected and subjected to modified Hematoxylin & Eosin staining as well as modified Masson’s trichrome staining. The tissues were evaluated and graded based on the characteristics of NAFLD (Kleiner et al., 2005; Brunt et al., 2011). For the statistical analysis, the livers were evaluated by two-way ANOVA, followed by the Tukey multiple comparison test. Results are given as the mean ± standard error of mean (S.E.M.). A value of p < 0.05 was considered as significant. The standard error of mean in each group has been updated in the table 1 recpectively. [page 9. Table 1]

Kleiner, D. E., Brunt, E. M., Van Natta, M., Behling, C., Contos, M. J., Cummings, O. W., ... & Nonalcoholic Steatohepatitis Clinical Research Network. Design and validation of a histological scoring system for nonalcoholic fatty liver disease. Hepatology. 2005, 41(6):1313-1321. doi:10.1002/hep.20701

Brunt, E. M., Kleiner, D. E., Wilson, L. A., Belt, P., Neuschwander‐Tetri, B. A., & NASH Clinical Research Network (CRN). Nonalcoholic fatty liver disease (NAFLD) activity score and the histopathologic diagnosis in NAFLD: distinct clinicopathologic meanings. Hepatology. 2011, 53(3):810-820. doi:10.1002/hep.24127

5) Analyses presented in Table 1 are not consistent with gene expression data in Fig. 4 which show no induction of inflammation and fibrosis with atherogenic diet.  This discrepancy calls the validity of the mouse model into question and needs to be addressed.

ANS: We would like to thank the reviewer for the feedback given. Recently, we tried to re-analyzed the gene expression associated with inflammation and fibrosis (Fn and Tnf-α) and linked it to the table 1. To confirm our data, we redo the relative gene expression analysis on inflammation- and fibrosis- related gene. We found that compared to the atherogenic diet group, the Tnf-α was significantly downregulated following the treatment with MERTK+/hi M2c macrophages. Fibrosis-related gene such as col1a1 and Fn were also downregulated compared to the atherogenic diet group. in parallel, NAS score also showed an overall decrease compared to the atherogenic diet group. The table 1 has been updated following reviewer’s advice. [page 9-10]

6) The authors' conclusion regarding increased HDL as a mechanism for improvement in NAFLD progression is problematic. Wildtype C57Bl6 mice, like other rodents, and unlike humans, carry the majority of serum cholesterol in HDL. An increase in HDL in this model indicates an increase in serum cholesterol (also indicated by the total serum cholesterol data in Fig. 2D), which is not an indicator of improved metabolic health.

ANS:

HDL is known as a negative predictor of NAFLD. In female C57Bl6 mice, the majority of serum HDL was decreased following the high-fat diet or atherogenic diet. Therefore, an increase in serum HDL might be a good sign in the treatment group. It also has been reported that HDL offers a hepatoprotective properties (Pierantonelli et al., 2020). In addition, the presence of high HDL may have a beneficial role in preventing atherosclerotic cardiovascular disease since NAFLD has been strongly linked to the obesity-related disease and CVD (Estes et al., 2018). A recent study conducted by Zhang et al. (2020) demonstrated that HFD rats showed an increase in the plasma lipid levels of total cholesterol (TC), triglyceride (TG) and low-density lipoprotein (LDL) levels. Furthermore, treatment with Corilagin (Cori) suppressed the plasma TC, TG, and LDL as well as significantly improving serum HDL-C relative to those of HFD group. While in our study, we also observed similar phenomenon where C57Bl/6 mice fed an atherogenic diet significantly downregulate serum HDL with increased level of LDL. In contrast, treatment group showed significant increase in serum HDL. In a study conducted by (Paigen et al., 1987), they demonstrated that C57BL6 mice fed a normal chow diet has a tota choesterol around 66±14 mg/dl, HDL 62±9 mg/dl, and LDL 4 mg/dl. Atherogenic diets further elevate the total cholestrol to 192±26 mg/dl, LDL 153 mg/dl, and lowering HDL 39±7 mg/dl. In parallel to our study, we found that the LDL level was higher in atherogenic diet group. As expected, the HDL was increased significantly following the treatment of MERTK+/hi M2 macrophages. Besides, the ratio of TC to HDL is a significant predictor of NAFLD. As the ratio increase, the NAFLD disease becomes more apparent. Nonetheless, HDL may have a hepatoprotection properties against NAFLD and other NAFLD-related disease. We thank the reviewer for the comment and we have added more information in the manuscript accordingly [Discussion section, page 13, line 588].

Pierantonelli, I., Lioci, G., Gurrado, F., Giordano, D. M., Rychlicki, C., Bocca, C., ... & Svegliati‐Baroni, G. (2020). HDL cholesterol protects from liver injury in mice with intestinal specific LXRα activation. Liver International, 40(12), 3127-3139.

Estes, C., Razavi, H., Loomba, R., Younossi, Z., & Sanyal, A. J. Modeling the epidemic of nonalcoholic fatty liver disease demonstrates an exponential increase in burden of disease. Hepatology. 2018, 67(1), 123-133.

Zhang, R., Chu, K., Zhao, N., Wu, J., Ma, L., Zhu, C., ... & Liao, M. (2020). Corilagin alleviates nonalcoholic fatty liver disease in high-fat diet-induced C57BL/6 mice by ameliorating oxidative stress and restoring autophagic flux. Frontiers in pharmacology, 10, 1693.

Paigen, B., Holmes, P. A., Mitchell, D., & Albee, D. (1987). Comparison of atherosclerotic lesions and HDL-lipid levels in male, female, and testosterone-treated female mice from strains C57BL/6, BALB/c, and C3H. Atherosclerosis, 64(2-3), 215-221.

7) What was the sex of the mice used? Male?

ANS: in this study, female C57BL6 mice were used.  

8) Overall, the data presented do not support the working model in Fig.6.

ANS: we thank the reviewer for the comment and major insight for this study. We have already made some changes regarding the data presentation and explained some major problems that should be further addressed. According to the reviewer’s suggestion, some changes in the manuscript have been implemented such as:

  1. Mice body weight analysis was updated.
  2. The aim of the study has been updated. In this context, our focus will be on how MERTK+/hi M2c macrophages, with enhanced reparative properties and cholesterol efflux, may contribute in alleviating NAFLD.
  3. Information regarding sham-retro orbital injection (100 μl of PBS) in the atherogenic diet group has been added into the manuscript.
  4. Statistics evaluation and table 1 has been added to the manuscript. Analyses of two-way ANOVA, followed by the Tukey multiple comparison test were conducted. Results are given as the mean ± standard error of mean (S.E.M.).
  5. Re-analysis of the gene related to inflammation and fibrosis were conducted. Therefore, the data on figure 4 has been updated which shown significant decrease in inflammation and fibrosis compared to the atherogenic diet.
  6. Figure 6 has been updated to the newest version.

Reviewer 2 Report

The manuscript : “MERTK +/hi M2c Macrophages Induced by Baicalin Alleviate Non-Alcoholic Fatty Liver Disease by Junior et al. (only one name listed)“ suggests that baicalin treated macrophages are so called M2c cells, and protect from NASH induced by an atherogenic diet, when injected retro-orbital for 3 times.

Explain abbreviations before used the first time.

Why were the cells not injected i.p.?

Do the macrophages indeed reach the liver?

A control group injected M-CSF differentiated macrophages is missing but is absolutely necessary.

Fibrosis staining is not convincing. Is alpha-SMA and collagen protein induced in the NASH liver, and do levels decline in the M2c treated animals?

Figure 5, which of the genes are characteristic for M2 cells?

“Recent study has shown that baicalin is capable of promoting M2 macrophages polarization during inflammation via up-regulation of IRF4 protein expression and down-regulation of IRF5, TNF-α, and IL-23 in vitro [16].” Please label these genes in figure 5.

How do the authors explain that mice fed the atherogenic diet have similar body weight at week 6 and that body weight declined in the treated group at week 8?

Was there a significant improvement of the NAS score?

What is shown in figure 4? Is this really delta Ct?

Author Response

# ijms-1377125

The authors thank the reviewers for spending considerable amount of time and efforts on examining this manuscript and greatly appreciate the constructive comments and the suggestions made by the reviewers. The authors have tried to respond to all the comments and accommodate most of the suggestions in the revised manuscript.

The authors’ answer:

        The authors thank and appreciate the reviewer’s comments. Therefore, we’ve revised our manuscript according to reviewer advises.

Reviewer #2

Comments and Suggestions for Authors

The manuscript: “MERTK +/hi M2c Macrophages Induced by Baicalin Alleviate Non-Alcoholic Fatty Liver Disease by Junior et al. (only one name listed)“ suggests that baicalin treated macrophages are so called M2c cells, and protect from NASH induced by an atherogenic diet, when injected retro-orbital for 3 times.

Explain abbreviations before used the first time.

ANS: We would like to thank the reviewer for the reminder. The manuscript has been updated accordingly.

Why were the cells not injected i.p.?

ANS: We would like to thank the reviewer for the feedback. To our opinion, intravenous injection (i.v.) is a suitable administration route for the cells to reach the liver. To support our opinion, a study conducted by Reyes et al. (2019) demonstrated that delivery of macrophages via intraperitoneal injection did not affect the outcome of dinitrobenzene sulphonic acid (DNBS)-induced colitis. In contrast, intravenous delivery of the macrophages reduced disease severity. Besides, intravenous but not intraperitoneal injection revealed greater accumulation of cells in lung, liver, and spleen which reduced disease severity. Moreover, a recent study from Lewis et al. (2020) also revealed that the Primary BMDMs localized to liver and spleen within 4 hours following intravenous injection in mice, which reduced the onset of hepatic necrosis. Based on these literatures, we are confident that intravenous injection is the suitable administration route for the macrophages to reach the target site. We have added more information regarding this matter in the manuscript on page 4, line 177.

Reyes, J. L., Lopes, F., Leung, G., Jayme, T. S., Matisz, C. E., Shute, A., ... & McKay, D. M. (2019). Macrophages treated with antigen from the tapeworm Hymenolepis diminuta condition CD25+ T cells to suppress colitis. The FASEB Journal, 33(4), 5676-5689.

Lewis, P. S., Campana, L., Aleksieva, N., Cartwright, J. A., Mackinnon, A., O'Duibhir, E., ... & Forbes, S. J. (2020). Alternatively activated macrophages promote resolution of necrosis following acute liver injury. Journal of hepatology, 73(2), 349-360.

Do the macrophages indeed reach the liver?

ANS:

We thank the reviewer for the questions. Indeed, we did not check whether the MERTK M2c macrophages reach the liver or not. This was also our concern at the beginning of the experiment. In this case, one of the approach is to use mouse bone-marrow derived macrophages (BMDM) transduced ex vivo with lentivectors expressing green fluorescent protein (GFP) driven by a synthetic promoter, attached it to the MERTK promotor or so, to track the macrophages pool in the recipient mice. However, we were concern that this cell modification could alter the nature of M2 macrophages and affect the out coming results. One of the solutions that we offer to answer this particular question is to analyze the MERTK level in the liver of each group since our M2c macrophages express high MERTK. Compared to the control group and untreated group, treatment group displayed elevated level of MERTK, suggesting MERTK M2 macrophages infiltration in the liver. The manuscript has been modified accordingly in page 9, line 335; figure 4.

A control group injected M-CSF differentiated macrophages is missing but is absolutely necessary.

ANS:

We would like to thank the reviewer for the comment. M2 macrophages can be further subdivided into M2a (wound healing macrophages), M2b (protective and pathogenic roles, and secrete both pro- and anti-inflammatory cytokines), M2c (regulatory phenotype, repress inflammation and fibrosis, promote tissue repair), and M2d (mainly contributing to angiogenesis and metastasis, supporting tumor growth) subtype. In addition, M2c subtype has been known to express higher MERTK receptor which demonstrates efferocytosis and phagocytosis capability, and enhanced cholesterol efflux with anti-inflammatory activity [19, 22, 23]. Based on the facts that NAFLD has a strong correlation with excess inflammation response and impaired cholesterol efflux capacity (CEC), we first confirm the efficacy of MERTK+/hi M2c macrophages to alleviate NAFLD first, and therefore it is not our focus of study to compare the efficacy of M2 vs M2c macrophages. To our knowledge, this is the first study that offers the M2c macrophages cell-based therapy to alleviate such disease. We have made an update in the introduction section. [page 2, line 63; page 3; line124].

Fibrosis staining is not convincing. Is alpha-SMA and collagen protein induced in the NASH liver, and do levels decline in the M2c treated animals?

ANS:

We thank the reviewer for the comment. Regarding fibrosis staining, modified Masson’s trichrome staining is frequently used in the clinical setting since its nature to stain the connective tissue. We totally agree with the reviewer that alpha-SMA and collagen protein staining are one of many approaches to mark the fibrosis, however, currently we did not offer the data. Besides, we also re-analyzed the gene expression in the livers in each group. Marker for fibrosis such as fibronectin (Fn) and collagen type 1 alpha (Col1a1) were used and compared with both control and untreated group (Figure 4). Compared to the untreated group, treatment with MERTK+/hi M2 macrophages decrease the overall fibrosis-related Col1a1 and Fn gene, suggesting suppression of fibrosis event in the liver. Page 10, figure 4.

Figure 5, which of the genes are characteristic for M2 cells?

ANS:

We would like to thank the reviewer’s feedback. The figure 5 displayed the difference genes between M-CSF induced M2 and M-CSF + baicalin-induced MERTK+/hi M2c subtype macrophages. In this context, we can show some M2c-related genes such as Thbs1 and Fpr1. The manuscript has been updated following the reviewer’s comment (Page 10).

“Recent study has shown that baicalin is capable of promoting M2 macrophages polarization during inflammation via up-regulation of IRF4 protein expression and down-regulation of IRF5, TNF-α, and IL-23 in vitro [16].” Please label these genes in figure 5.

ANS:

We thank the reviewer for the comment. This statement originally came from a paper stated that baicalin could promote repolarization from LPS-induced M1 macrophages to M2 phenotypes with upregulation of IRF4 protein expression and down-regulation of IRF5, TNF-α, and IL-23 in vitro. In our case, it’s a whole different experiment setting. We use mononuclear cells (MNCs)-derived M2 macrophages induced by M-CSF and further polarize it into MERTK+/hi M2c subtype with baicalin. However, we could label other genes related to the M2 macrophages such as Vegfa, Rsc1a1, Thbs1, Aoah, Ccl6, etc. We have made a change in the figure 5 and have labelled the genes of M2 macrophages.

How do the authors explain that mice fed the atherogenic diet have similar body weight at week 6 and that body weight declined in the treated group at week 8?

ANS:

We would like to thank the reviewer for the feedback regarding the body weight of the mice. To answer this particular question, we changing the way of explaining the figure. After injection, the M2c group mice were not shown a reduction in body weight, which was not significant compared to the control group from the fourth week to the seventh week. Compared with the M2c group, the Ath group showed a significant difference in body weight only at the fifth week (P<0.0393) and the seventh week (P<0.0497) after injection. However, the body weight at sixth week (mean±sem: 21.3±0.45) showed not significant compared to the fifth week (22.19±0.59) and the seventh week (21.88±0.59). Compared with M2c group, the body weight in ath diet group after injection are overall increased. The manuscript has been updated rescpectively (Result section 2.2, page 5)

Was there a significant improvement of the NAS score?

ANS:

Statistics regarding NAFLD scoring (table 1) has been added and updated. Based on the evaluation and grading of NAFLD in each group, there was significant improvement of the NAS score in the treatment group compared to the mice. The manuscript has been updated accordingly (Result section 2.3, page 9, table 1).

What is shown in figure 4? Is this really delta Ct?

ANS:

We would like to thank the reviewer’s suggestion and careful review. We re-analyze and re-examine the relative gene expression data again using the standardized mRNA level comparative methods 2−ΔΔCt. The relative gene expression of target genes was normalized to β-actin (housekeeping gene). We also update the latest methods and results in the manuscript (page. 16 and 10 (figure 4), respectively)

Round 2

Reviewer 1 Report

The authors have made several improvements to the manuscript, which have addressed several points raised during the initial review.  However, some major points remain:

1) Given that the authors acknowledged that food intake was not assessed and body weight was reduced in the M2c treatment group, these points need to be clearly addressed in the Discussion section of the manuscript.  In particular, the contribution of decreased body weight to NAFLD progression needs to be acknowledged in relation to the effects observed in the M2c group.  

2) Although the authors have added an additional reference to support their conclusion regarding the contribution of increased HDL to the observed improvement in NAFLD with M2c treatment, the fact remains that in wild type mouse strains such as C57BL6, the majority of cholesterol is carried in HDL.  This raises questions about the data presented in Fig. 2D and any conclusions based on this data. 

3) Fig. 6 needs to be clearly described as a working model for how M2c macrophages may impact NAFLD.

Author Response

# ijms-1377125

The authors thank the reviewers for spending considerable amount of time and efforts on examining this manuscript an d greatly appreciate the constructive comments and the suggestions made by the reviewers. The authors have tried to respond to all the comments and accommodate most of the suggestions in the revised manuscript.

Reviewer #1

Comments and Suggestions for Authors

The authors have made several improvements to the manuscript, which have addressed several points raised during the initial review.  However, some major points remain:

Point 1: Given that the authors acknowledged that food intake was not assessed and body weight was reduced in the M2c treatment group, these points need to be clearly addressed in the Discussion section of the manuscript.  In particular, the contribution of decreased body weight to NAFLD progression needs to be acknowledged in relation to the effects observed in the M2c group.  

Response 1: We would like to thank the reviewer for the meticulous feedback regarding the reduction of body weight in the treatment group of M2c compared to the other group with sham treatment. At present, we already made some corrections and descriptions regarding this matter in the discussion section. We have added a new supporting reference in relation to the effect of body weight loss in NAFLD and its beneficial impact. Please kindly refers to page 13, discussion section, 2nd paragraph, line 438 (PDF file).

Mota-Rojas, D., Olmos-Hernández, A., Verduzco-Mendoza, A., Hernández, E., Martínez-Burnes, J., & Whittaker, A. L. (2020). The utility of grimace scales for practical pain assessment in laboratory animals. Animals10(10), 1838.

Point 2: Although the authors have added an additional reference to support their conclusion regarding the contribution of increased HDL to the observed improvement in NAFLD with M2c treatment, the fact remains that in wild type mouse strains such as C57BL6, the majority of cholesterol is carried in HDL.  This raises questions about the data presented in Fig. 2D and any conclusions based on this data. 

Response 2: We are thankful that the reviewer has raised an important point here regarding the fact that most of the cholesterol in wild type mouse strains is carried in HDL. Therefore, we have added this statement in our manuscript along with its reference. Please kindly refers to result section 2.2, page 5, line 201 (PDF file).

Camus, M. C., Chapman, M. J., Forgez, P., & Laplaud, P. M. (1983). Distribution and characterization of the serum lipoproteins and apoproteins in the mouse, Mus musculus. Journal of lipid research24(9), 1210-1228.

Point 3: Fig. 6 needs to be clearly described as a working model for how M2c macrophages may impact NAFLD.

Response 3: We agree with the reviewer’s suggestion on a clear description of a working model for how M2c macrophages may impact NAFLD and have incorporated the suggestion throughout the manuscript. We tried to modify the figure 6 as suggested and carefully described M2c macrophages as the working model in detail in page 16, figure 7.

Please kindly see the attachment for the detail. 

Reviewer 2 Report

The authors may consider to use a different housekeeper for real-time RT-PCR. The beta-actin genes is highly expressed compared to the further genes analyzed. It is also more convenient to show gene expression instead of CT values. 

The staining for fibrosis has to be improved or immunoblots analyzing fibrosis-specific proteins have to be provided. 

Author Response

# ijms-1377125

The authors thank the reviewers for spending considerable amount of time and efforts on examining this manuscript an d greatly appreciate the constructive comments and the suggestions made by the reviewers. The authors have tried to respond to all the comments and accommodate most of the suggestions in the revised manuscript.

Reviewer #2

Point 1: The authors may consider to use a different housekeeper for real-time RT-PCR. The beta-actin genes are highly expressed compared to the further genes analyzed. It is also more convenient to show gene expression instead of CT values. 

Response 1: We would like to thank the reviewer for the insight. However, in this study, beta-actin was used as the housekeeping gene. Based on Araujo et al., the constitutive expression of β-actin gene level was not altered in relation to control (normal diet mice vs. high fat diet mice). We therefore added this reference (80) in our manuscript to manifest the housekeeping gene as the absolute gene expression level in the data. The formula 2-∆∆CT was used as the way to measure the relative gene expression by subtracting the Ct value of target gene with CT value of housekeeping gene (in this setting, the beta-actin). Therefore, the relative gene expression of housekeeping gene can only be displayed as CT values. But all the genes that we measured were shown in gene expression (2-∆∆CT). In addition, the expression level of beta-actin (Ct values), was relatively the same, assuming the role of beta-actin as the house-keeping gene was indeed accurate and precise. Please kindly refers to material method section 4.7. page 19, line 777 (PDF file).

Araujo, L. C., Bordin, S., & Carvalho, C. R. (2020). Reference gene and protein expression levels in two different NAFLD mouse models. Gastroenterology research and practice, 2020.

Point 2: The staining for fibrosis has to be improved or immunoblots analyzing fibrosis-specific proteins have to be provided. 

Response 2: We would like to thank the assessment from the reviewer. We agree that the improvement of the staining for fibrosis is an important consideration. Therefore, we have incorporated more fibrosis staining sections. Please kindly refers to figure 4 and its legends we have corrected on page 9. We also modified our description about the fibrosis in the result section 2.3 on page 7.

Please kindly see the attachment for the detail.

Round 3

Reviewer 2 Report

Authors have adressed my concerns.